# OpenReview forum: "Cascade Reward Sampling for Efficient Decoding-Time Alignment"
_colmweb.org/COLM/2025/Conference — COLM 2025_

### Official Review · Reviewer_t2yh · 2025-04-12

**Rating:** 6
**Confidence:** 2
**Ethics Flag:** 1

**Summary:**

This paper introduces Cascade Reward Sampling (CARDS), a decoding-time alignment method that improves efficiency and performance by sampling and evaluating small semantic segments. The proposed method reduces redundant computation in both the LLM and the reward model (RM) via an uncertainty-based segmentation strategy. The authors demonstrate improved decoding speed and reward quality across several benchmarks.

**Reasons To Accept:**

1. Introduces a novel segment-level rejection sampling algorithm that is simple yet effective, achieving ~70% reduction in decoding time while maintaining or improving alignment.
2. The uncertainty-based segmentation mechanism is intuitive, practical, and aligns with the LLM’s internal understanding of generation.
3. Demonstrates substantial empirical improvements over strong baselines such as RAD/ARGS and naive rejection sampling.

**Reasons To Reject:**

1. The paper hinges on the idea that reward models behave like value functions over semantically complete prefixes, but this is neither theoretically proven nor empirically well-supported. Several derivations are internally inconsistent or depend on circular assumptions. For example, Equation 4 defines the value as an expectation over future tokens based on rewards that depend on those tokens, while Lemma 1 assumes the reward is already equal to that value.
2. Core claims regarding the use of reward scores for incomplete sequences appear speculative and are not sufficiently substantiated by prior literature.
3. The experiments primarily target a single dataset (HH-RLHF) and models with relatively modest scale. It is unclear whether the proposed approach holds under more diverse conditions or for larger-scale models.
4. Although segment-based rejection sampling improves per-token efficiency, the potential computational burden under high rejection rates or large batches is not addressed.
5. The manuscript introduces important assumptions and designs, such as partial reward estimation, semantic segmentation of generation, but lack of eferencing relevant prior work([1], [2], [3]) that has explored similar ideas.

[1] From r to Q∗: Your Language Model is Secretly a Q-Function, https://arxiv.org/abs/2404.12358
[2] PAD: Personalized Alignment at Decoding-Time, https://arxiv.org/abs/2410.04070
[3] Controlled Decoding from Language Models, https://arxiv.org/abs/2310.17022

---

> ### Author Response · Authors · 2025-06-02
>
> Thank you for raising your concerns to this paper. We would like to clarify our manuscript and address your questions in the responses below.
>
> **Q1:  Lemma 1 assumes the reward is already equal to that value.**
>
> A1: Thank you for your comment. **Our paper does not contain a “Lemma 1” and our Equation 4 refers to the target distribution, which is unrelated to value functions.** If you are referring to another work or a different section, please clarify, and we will be happy to address it directly.
>
> Regarding the relationship between RMs and value functions, we discuss their connection in **Section 4.3.3**, under uncertainty-based segmentation. We claim that RMs and value functions share similarity based on the empirical results of reward accuracy (**Fig. 2 and 3**).
>
> **Q2: Core claims regarding the use of reward scores for incomplete sequences are not sufficiently substantiated by prior literature.**
>
> A2: Thank you for highlighting this concern. Prior works have used RMs to score incomplete sequences in autoregressive generation [1, 2, 3], but these approaches lack justification for the accuracy of RMs on partial sequences. To address this gap, our paper provides new empirical evidence demonstrating that RMs can be accurate on semantically complete prefixes (see **Section 4.3**), clarifying the conditions under which RMs may be reliably applied to incomplete sequences. We will further clarify this contribution and its relation to prior work in the final version.
>
> [1] ARGS: Alignment as Reward-Guided Search. ICLR 2024.
>
> [2] RAIN: Your Language Models Can Align Themselves without Finetuning. ICLR 2024.
>
> [3] Reward-Augmented Decoding: Efficient Controlled Text Generation With a Unidirectional Reward Model. EMNLP 2023.
>
> **Q3: The experiments primarily target a single dataset (HH-RLHF) and models with relatively modest scale.**
>
> A3: Besides HH-RLHF, we also have empirical results on AdvBench, SafeRLHF, AlpacaEval 2.0, Ultrafeedback, BeaverTails, and HelpSteer. We emphasize the results on HH-RLHF because our RMs are trained on this dataset, but our results demonstrate strong generalization across diverse tasks and domains.
>
> Regarding model scale, we primarily use 7B models due to computational constraints, which is consistent with recent work [1, 4]. Scaling to larger models is an important direction, and we plan to explore this in future work.
>
> [4] TreeBoN: Enhancing Inference-Time Alignment with Speculative Tree-Search and Best-of-N Sampling. 2024.
>
> **Q4: The potential computational burden under high rejection rates or large batches is not addressed.**
>
> A4: Thank you for raising this important point. The computational burden under high rejection rates is primarily influenced by the reward threshold: lower thresholds reduce rejections and thus computational cost. For batched inference, we describe a simple batching strategy in **Appendix A.2**, which allows larger batches to be processed more efficiently than smaller ones.
>
> **Q5: Lacking referencing relevant prior works.**
>
> A5: Thank you for pointing this out. While we briefly mention controlled decoding in **L92**, we agree that the prior works you referenced [5, 6, 7] are relevant. They share a similar alignment framework of reward-guided generation, but focus more on the design of the reward signals rather than the generation strategy as studied in our paper. We will update the related work section to better contextualize our contributions within this literature.
>
> [5] From r to Q∗: Your Language Model is Secretly a Q-Function. COLM 2024.
>
> [6] PAD: Personalized Alignment at Decoding-Time. ICLR 2025.
>
> [7] Controlled Decoding from Language Models. ICML 2025.

---

> > ### Comment · Reviewer_t2yh · 2025-06-10
> > **Thanks for the reply.**
> >
> > Thanks for the authors' reply. Most of my concerns have been solved, so I will raise my score.

---

> ### Author Response · Authors · 2025-06-08
>
> Dear Reviewer,
>
> Thank you for your valuable feedback and engagement throughout the review process. We want to remind you that the deadline for the author-reviewer discussion period is approaching.
>
> We have not yet received a response to our posted replies, which we believe address the concerns you raised regarding our paper. We kindly encourage you to review our responses and, if you find them satisfactory, to reconsider your rating.
>
> Authors

---

> ### Comment · Area_Chair_3Q1m · 2025-06-09
> **Please respond to the reviewers**
>
> Reviewer t2yh, please consider the authors' response and the other reviews.
> Thanks,
> Area chair

---

### Official Review · Reviewer_rS2s · 2025-05-12

**Rating:** 8
**Confidence:** 4
**Ethics Flag:** 1

**Summary:**

The paper addresses the problem of efficient decoding-time LLM alignment via a segment-level rejection sampling algorithm. Rather than using fixed-length segments, it employs an uncertainty-based segmentation mechanism that hypothesizes segment boundaries when the entropy of the next-token probability distribution crosses a threshold. Rejection sampling is then performed at the segment level, with hyperparameter settings changing dynamically to bias towards high-reward prefixes. Hypotheses are scored by item-level rewards models. The authors compare the accuracy of reward models on incomplete text against fixed-length segmentation  and demonstrate that uncertainty-based segmentation has the higher correlation with reward model behavior on full text. The authors evaluate on three different standard alignment benchmarks and achieve efficiency gains of ~70% over a range of baselines, and win-tie rates > 90%, topping all competing approaches in most cases.

**Questions To Authors:**

In general, the segmentation into semantically coherent units as determined by entropy of next-token probability distribution may not be equally helpful for all alignment dimensions - i.e., some segmentations (longer ones?) might be more suitable for 'helpfulness' as opposed to 'safety'. I didn't see a discussion of this in the paper - do e.g., the hyperparameters affecting segmentation play out differently on different benchmarks highlighting different alignment dimensions?

**Reasons To Accept:**

The paper proposes a simple but effective way of addressing the problem of segment-level reward evaluation (uncertainty-based segmentation plus rejection sampling), which previous solutions have not addressed in a satisfactory manner. The correlation analysis between full-text reward and segment-level reward clearly demonstrates the advantage of uncertainty-based sampling.
The experimental results are convincing - I appreciate the comprehensiveness of the experiments (number of benchmarks and baselines) as well as the detailed ablation studies in the Appendix. Results are convincing.
The prior work section is sufficient - I am not aware of  any major gaps in the comparison to previous work.

**Reasons To Reject:**

I don't see any reasons to not publish.

---

> ### Author Response · Authors · 2025-06-02
>
> We truly appreciate your positive feedback and the acknowledgment of our experiments. We would like to address your questions in the response below.
>
> **Q1: Some segmentations (longer ones?) might be more suitable for 'helpfulness' as opposed to 'safety'.**
>
> A1: Thank you for raising this interesting point. We agree that optimal segment length may vary for different objectives. Our segmentation hyperparameters for all experiments were originally tuned on HH-RLHF for harmlessness & helpfulness (examples in HH-RLHF mainly focus on safety), resulting in relatively short segments (7~8 tokens on average). To explore the impact on helpfulness, we used general utility as the evaluation metric. We increased the uncertainty threshold to 4.0 (yielding segments of ~20 tokens on average) and present the results below, evaluated on the same data as Table 4 and using llama-7b as the base model. The findings show that longer segments slightly improve general utility and diversity metrics. We will include this ablation study in the final version.
>
> | Uncertainty Threshold | Diversity | Coherence | LC Win Rate (%) | Win Rate (%) |
> |:---:|:---:|:---:|:---:|:---:|
> | 3.0 | 0.742 | 0.856 | 1.609 | 0.878 |
> | 4.0 | 0.863 | 0.818 | 1.714 | 0.881 |

---

### Official Review · Reviewer_Fnvn · 2025-05-13

**Rating:** 7
**Confidence:** 3
**Ethics Flag:** 1

**Summary:**

This paper presents a segment-level rejection sampling method called cascade reward sampling to improve the efficiency of decoding-time alignment. The proposed method has a granularity between token-level rejection sampling and item-level rejection sampling, and thus it can balance between the computation costs of LLM and reward model. Internally, the proposed method uses uncertainty as a measure to accurately and efficiently determine the segment of dynamic length, and empirical analyses are conducted to show that reward model on dynamic segments can achieve good performance similar to reward models on full sequence. Further evaluations are performed to highlight the effectiveness of the proposed method in efficiency and quality.

**Questions To Authors:**

- Line 164: it would be good to see how frequently this length limit is reached. Or even better, show some statistics like the distribution of segment length and the number of segments in one sequence.
- Sec 4.3.3: any empirical support for this claim?
- Table 2 and Table 3: why would the proposed method outperform item-level RS? Item-level RS uses RM on full sequence and should be more accurate?

**Reasons To Accept:**

- The proposed method tackles an important problem of efficient decoding-time alignment. It can be widely applied to many cases where reject sampling is needed.
- The empirical evaluation show significant improvements in efficiency and consistent quality wins against baselines.

**Reasons To Reject:**

- It is claimed that dynamic-length segmentation based on uncertainty is the key component of the proposed method. I would like to see more analyses/evidences on this. It might be useful to include fixed segmentation as part of the baselines in the evaluations.

---

> ### Author Response · Authors · 2025-06-02
>
> Thank you for your constructive feedback and for recommending acceptance of our paper. We appreciate your thoughtful comments and would like to clarify the points you raised in the review.
>
> **Q1: It might be useful to include fixed segmentation as part of the baselines in the evaluations.**
>
> A1: Thank you for this insightful suggestion. We have now included CARDS with fixed-length segmentation (we set the length to be 10 tokens, which approximates the average length from uncertainty-based segmentation) as a baseline in our evaluation (see table below). As shown, fixed segmentation results in lower alignment quality (RM Score: 7.84 vs. 8.30) and increased computational cost (Total Calls: 1058.33 vs. 872.91), confirming its limitations compared to our approach. The suboptimal performance originates from the incompleteness of these segments, which causes the partial reward scores to be no longer accurate. We will add these results to the final version.
>
> | Segmentation | RM Score | # LLM Calls | # RM Calls | # Total Calls | Total Calls (min) |
> |:---:|:---:|:---:|:---:|:---:|:---:|
> | US | 8.30 | 833.42 | 39.49 | 872.91 | 75.8 |
> | Fixed | 7.84 | 1032.56 | 25.77 | 1058.33 | 97.2 |
>
> **Q2: Line 164: it would be good to see how frequently this length limit is reached. Or even better, show some statistics like the distribution of segment length and the number of segments in one sequence.**
>
> A2: Thank you for raising this question. We already had visualizations of segment length statistics in **Fig. 7 and 8**. To accurately reflect the segmentation behavior, we did not enforce the 32-token length limit in these analyses. The results show that, under our reward threshold (3.0), the average segment length is 7~8 tokens, with most segments under 20 tokens. We will clarify these findings in the final version.
>
> **Q3: Sec 4.3.3: any empirical support for this claim?**
>
> A3: Thank you for highlighting this point. Our claim regarding the relationship between RMs and value functions is motivated by our observation that semantically complete prefixes preserve the accuracy of RMs. However, obtaining an accurate value function for direct empirical comparison remains challenging, as prior works [1, 2] rely on approximations and do not open-source their code. We will clarify this limitation in the final version and more explicitly ground our claim in the recent literature.
>
> [1] From r to Q∗: Your Language Model is Secretly a Q-Function. COLM 2024.
>
> [2] Controlled Decoding from Language Models. ICML 2024.
>
> **Q4: Table 2 and Table 3: why would the proposed method outperform item-level RS?**
>
> A4: Thank you for your question. It is true that RM on full sequence is more accurate. However, Item-level RS is less efficient than CARDS. Given a practical compute budget, CARDS identifies more aligned responses than item-level RS and therefore outperforms it in experiments. CARDS is more efficient because it generates responses segment by segment, conditioning each new segment on previously identified high-reward prefixes. This approach focuses the search on promising regions of the response space, increasing the likelihood of producing high-reward outputs compared to sampling entire responses independently, as in item-level RS. We will clarify this advantage in the revised manuscript.

---

> ### Author Response · Authors · 2025-06-08
>
> Dear Reviewer,
>
> Thank you for your valuable feedback and engagement throughout the review process. We want to remind you that the deadline for the author-reviewer discussion period is approaching.
>
> We have not yet received a response to our posted replies, which we believe address the concerns you raised regarding our paper. We kindly encourage you to have a look at our responses to help all reviewers move closer to consensus on this paper
>
> Authors

---

> > ### Comment · Reviewer_Fnvn · 2025-06-09
> >
> > Thanks for the additional information!

---

### Official Review · Reviewer_ToeH · 2025-05-13

**Rating:** 5
**Confidence:** 3
**Ethics Flag:** 1

**Summary:**

CARDS is a decoding-time alignment algorithm that produces one small segment of text at a time via rejection sampling. Specifically, it repeatedly samples segments of text (stopping right before a high-entropy prediction) and evaluates each with an RM until a candidate is accepted. The accepted segment is merged into the left context, and the sampling of the next segment begins. The reward threshold (which controls the probability of a segment being accepted) increases with sequence length as longer prefixes tend to have higher rewards.

The authors show that the rewards on uncertainty-based segments are highly correlated with those on complete responses, meaning that existing “item-level” RMs can be applied directly to their setup. They argue that CARDS is more efficient than other decoding-time alignment algorithms, due to balancing RM and LLM calls. Finally, in the main experiments, they show that CARDS outperforms a large set of baselines, including training with PPO/DPO, and other decoding-time algorithms like BoN and RAIN, when evaluated on datasets like HH-RLHF, AdvBench, and SafeRLHF.

**Questions To Authors:**

- In Table 4, how is it possible that the win rate for Llama + CARDS is 0.878, but the win rate for Mistral + CARDS is 3.445? Are they both percentages, i.e., 0.878% and 3.445%?

**Reasons To Accept:**

The paper is well-written. The approach is simple, and strikes a balance between approaches that require decoding the entire response before scoring (e.g., best-of-$n$) and token-level approaches that invoke a RM at every time step (e.g., RAD). When using both a RM and LLM-as-a-judge, CARDS outperforms a large set of baselines.

**Reasons To Reject:**

1. The experimental sections (§4 and §5) are missing basic details about the experimental setup in order to understand the results. For instance,
    - In both sections, which RM was used? Does the reward correlation between segments and full-length responses (§4.3.2) hold for multiple choices of RMs?
    - Does the efficiency analysis in Table 1 depend on hyperparameter choices? If so, how did you pick them? E.g., the efficiency of BoN must depend on the choice of $n$, and the efficiency of CARDS should depend on the choice of $r^\star$ (the final goal reward score) which affects the number of resampling iterations that would be required.
    - In §4.3, what choice of fixed length do you use for the comparison to fixed-length segments (Figures 2, 3)?
    - In §5, what data was used for training the PPO and DPO baselines?
    - For CARDS, what hyperparameters do you choose for $\tau_u$ (the entropy threshold) and $\alpha$, $r^\star$, $\beta$ used in rejection sampling? What value of $n$ did you use for best of $n$? Did you do a hyperparameter search for each baseline?
    - In §5.3, how do you measure the diversity and coherence of responses?

2. Most of the results in the paper use LMs to assign scores from 1-5. This is very prone to biases, no matter how well you optimize the prompts, and the numbers in Tables 2 and 4 are all incredibly close. It would be helpful to report the standard deviation and/or statistical significance tests. Head-to-head comparisons are more reliable, but I am confused about the win-tie % metric used in Table 3. If A > B 5% of the time, B > A 10% of the time, and A = B the remaining 85% of the time, does that mean that the win-tie % for A would be 90% (despite actually being the worse method)?

3. The main novelty of CARDS is performing rejection sampling on one segment of text at a time, with an entropy-based threshold for determining segment boundaries. It will be helpful to compare to an ablation that uses sentence boundaries instead of entropy-based segments. The related work should also give more discussion to process reward modeling, which evaluates the quality of each individual step and shares a lot of similarity with CARDS.

Small suggestions:

- Align decimal numbers in Table 1 so that the decimal point lies on the same vertical axis; it will bring out the differences in order of magnitude.
- Include arrows next to the metrics (e.g., ASR ($\uparrow$)) as there is a mix of both higher-is-better and lower-is-better.

---

> ### Author Response · Authors · 2025-06-02
>
> Thanks for your detailed and thoughtful feedback, as well as for recognizing the strengths of our method. We appreciate the opportunity to address your concerns and clarify several points that may have led to misunderstandings in our manuscript. We hope that our responses below will clarify the value and rigor of our work.
>
> **Q1: In §4 and §5, which RM is used? Does the reward correlation between segments and full-length responses (§4.3.2) hold for multiple choices of RMs?**
>
> A1: As noted in **L217**, our main experiments use a RM trained from Llama-7B on HH-RLHF. To assess the robustness of our findings, we also evaluated reward correlation using an RM trained from Llama-2-7B on UltraFeedback, as shown in **Fig. 13**. The reward correlation between segments and full-length responses remains consistent across these different RMs, demonstrating that our results generalize well to multiple RM architectures and datasets.
>
> **Q2: Does the efficiency analysis in Table 1 depend on hyperparameter choices? If so, how did you pick them?**
>
> A2: The efficiency analysis in Table 1 does depend on hyperparameter choices. For CARDS, we performed a grid search to select hyperparameters, as detailed in **Appendix B.3**. For RAD/ARGS and TreeBoN, we adopted the recommended settings from their code for a fair comparison. For item-level RS, we set the reward threshold to match CARDS (r=8.5), and for BoN, we used the same number of samples as ARGS (n=20). These choices ensure a fair comparison between baselines.
>
> **Q3: In §4.3, what choice of fixed length do you use for the comparison to fixed-length segments (Fig. 2, 3)?**
>
> A3: For the fixed-length segment comparisons in Fig. 2 and 3, we use the 25%, 50%, and 75% prefixes of the full responses and report the average reward score across these segments. We appreciate your suggestion and will include this clarification in the final version.
>
> **Q4: In §5, what data was used for training the PPO and DPO baselines?**
>
> A4: Both the DPO and PPO baselines are trained on the HH-RLHF dataset, using both Llama-7B and Mistral-7B-v0.2 as base models. This ensures a fair and consistent comparison across methods. We will release the model weights upon acceptance.
>
> **Q5: What hyperparameters are used for CARDS ($\tau_u$, $\alpha$, $r^\star$, $\beta$)? What value of N is used for BoN?**
>
> A5: The hyperparameter values for CARDS ($\tau_u$, $\alpha$, $r^\star$, $\beta$) are summarized in **Appendix B.3**. These were selected via grid search to ensure optimal performance. For BoN, we set $n=20$, following the recommendation in [1]. We will ensure these details are presented in the final version.
>
> [1] https://github.com/deeplearning-wisc/args
>
> **Q6: In §5.3, how do you measure the diversity and coherence of responses?**
>
> A6: As described in **L282**, we follow the evaluation of [2] (Section 3.1) to measure diversity and coherence. Specifically, diversity is the aggregation of n-gram repetition rate, while coherence is the cosine similarity between the sentence embeddings of the prompt and its continuation. We will clarify these definitions in the final version.
>
> [2] ARGS: Alignment as Reward-Guided Search. ICLR 2024.
>
> **Q7: Most of the results in the paper use LMs to assign scores from 1-5. This is very prone to biases.**
>
> A7: This evaluation setup follows previous works [2, 4], and we believe it is fair if we compare the scores of different methods on the same set of prompts. We agree that a head-to-head comparison might be more unbiased, and this is precisely why we included Table 3.
>
> **Q8: It would be helpful to report the standard deviation and/or statistical significance tests.**
>
> A8: Thank you for your suggestion. Due to computational constraints–primarily the use of 7B models and the HH-RLHF dataset (12.5k rows)--we were unable to run repeated experiments to report standard deviations or statistical significance tests. This is consistent with prior works [2, 3, 4], which also do not report error bars or significance tests. We agree that including such analysis would strengthen the results, and we will consider this in future work as resources allow.
>
> [3] RAIN: Your Language Models Can Align Themselves without Finetuning. ICLR 2024.
>
> [4] Reward-Augmented Decoding: Efficient Controlled Text Generation With a Unidirectional Reward Model. EMNLP 2023.

---

> > ### Author Response · Authors · 2025-06-02
> >
> > **Q9: I am confused about the win-tie metric used in Table 3.**
> >
> > A9: We follow previous work [2] to report the win-tie rate. To address your concern, we provide the detailed win-tie-loss rates for Llama-7B on HH-RLHF in the table below. As shown, CARDS consistently outperforms baselines across most settings, demonstrating its strong alignment quality.
> >
> > | Compared Method | GPT-4 win | GPT-4 tie | GPT-4 lose | Claude-3 win | Claude-3 tie | Claude-3 lose |
> > |:---:|:---:|:---:|:---:|:---:|:---:|:---:|
> > | Vinalla LLM | 0.43 | 0.56 | 0.01 | 0.37 | 0.59 | 0.04 |
> > | PPO | 0.38 | 0.26 | 0.36 | 0.34 | 0.26 | 0.40 |
> > | DPO | 0.38 | 0.41 | 0.21 | 0.40 | 0.43 | 0.17 |
> > | ARGS | 0.43 | 0.30 | 0.27 | 0.30 | 0.42 | 0.28 |
> > | RAIN | 0.24 | 0.72 | 0.04 | 0.29 | 0.56 | 0.15 |
> >
> > **Q10: It will be helpful to compare to an ablation that uses sentence boundaries instead of entropy-based segments.**
> >
> > A10: We already included this ablation in **Appendix D.4**, where we compare entropy-based segmentation with sentence boundary segmentation. Our results show that using sentence boundaries often leads to longer segments, which reduces efficiency compared to our entropy-based approach.
> >
> > **Q11: The related work should also give more discussion to process reward modeling.**
> >
> > A11: Thank you for this valuable suggestion. While PRMs share similarities with how CARDS utilizes RMs, as noted in **L111**, PRMs are primarily designed for different tasks. We agree that integrating PRMs as reward signals could further enhance CARDS, especially for reasoning tasks, and we will discuss this direction in the revised related work section. Exploring this integration is an exciting avenue for future research.
> >
> > **Q12: Small suggestions on decimal numbers and metrics.**
> >
> > A12: Thank you for pointing this out. These typos will be revised in the final version.
> >
> > **Q13: In Table 4, how is it possible that the win rate for Llama + CARDS is 0.878, but the win rate for Mistral + CARDS is 3.445?**
> >
> > A13: Thank you for pointing out this potential confusion. The win rates in Table 4 are reported as percentages, so 0.878 and 3.445 correspond to 0.878% and 3.445%, respectively. These low values are due to the evaluation setting, where Llama/Mistral (old 7B models) are compared against the newest GPT-4-turbo. These results are consistent with [5], which also achieves ~3% win rates on AlpacaEval 2.0 with 7B/13B models. We will clarify the units in the table to avoid any misunderstanding in the final version.
> >
> > [5] QLORA: Efficient Finetuning of Quantized LLMs. NeurIPS 2023.

---

> > ### Comment · Reviewer_ToeH · 2025-06-09
> >
> > Thank you for the detailed responses! I want to follow up on a few of them below.
> >
> > **Q2:** I'm confused why it makes sense to compare the efficiencies of different methods in a vacuum; since these are methods of scaling the test-time compute, the efficiency depends crucially on the choice of parameters. For instance, CARDS is more efficient than BoN with $n=20$, but it looks like it would be less efficient for BoN $n=6$. In particular, TreeBoN achieves really similar efficiency to CARDS, so it is possible that a different choice of hyperparameters would change the relative efficiencies. What we really want to see how is how performance scales with compute for each of these methods, in order to truly make the argument that CARDS can achieve better performance at a lower cost.
> >
> > **Q3:** What happens if you use the best of 25%, 50%, and 75% segments instead of averaging across the three? For CARDS, do you also use the first uncertainty-based segment, or do you average across all uncertainty-based segments? If the latter, what happens if you e.g., average across the four 25% segments?
> >
> > **Q5**: The authors mention that a hyperparameter search was performed for CARDS, but defaults were used for all the baselines. This is not a fair comparison, because it means that only CARDS had hyperparameters optimized for this experimental setup. In particular, Tables 7 and 8 in the appendix show that different CARDS hyperparameters were even used for `Llama-7B` and `Mistral-7B`, whereas I assume the same hyperparameters were used across these two models for all baselines.
> >
> > **Q8:** Reporting standard deviation or performing statistical significance tests does not require any additional compute, and would make the comparison of such close numbers more grounded. Prior work not doing so is not a good reason to not do so in this work. In addition, the win/tie/lose rates for **Q9** show that CARDS actually matches or underperforms PPO.
> >
> > **Q10:** In Appendix D.4, we see that sentence-level segmentation outperforms the uncertainty-based segmentation used in CARDS at some cost to efficiency. Is it possible to modify the hyperparameters in the sentence-level case, e.g., by lowering target reward $r^\star$ or using more lenient rejection sampling hyperparameters ($\alpha$ and $\beta$) to match the performance of uncertainty-based segmentation without any increase in inference time?

---

> > > ### Author Response · Authors · 2025-06-10
> > >
> > > **Q2**
> > >
> > > A2: For all test-time scaling baselines, it is essential to prioritize alignment quality when selecting hyperparameters, before considering computational efficiency. In our experiments, we follow the recommended settings from prior work to ensure that each baseline operates in its intended regime — settings that are known to provide good alignment. While adjusting these hyperparameters could potentially enhance efficiency, such changes often compromise alignment quality. As the primary objective of test-time scaling methods is to maintain alignment, optimizing purely for efficiency falls outside the intended scope of these approaches. Given this, we believe our comparison offers a fair and meaningful evaluation.
> > >
> > > **Q3:**
> > >
> > > A3: If you are referring to the reward accuracy comparison in Fig. 2b, we can show the requested “best segment” results in the following table. Our proposed uncertainty-based segmentation still gets lower rewards for rejected responses, which is aligned with the results in Fig. 2b.
> > >
> > > | **Segmentation** | **$\tau_u=0$** | **$\tau_u=2$** | **$\tau_u=4$** | **$\tau_u=6$** | **$\tau_u=8$** |
> > > |:---:|:---:|:---:|:---:|:---:|:---:|
> > > | US + IR (ours) | 3.96 | 3.79 | 3.56 | 3.78 | 3.95 |
> > > | Fixed + IR | 3.98 | 4.03 | 4.06 | 4.01 | 3.97 |
> > >
> > > The CARDS framework sequentially samples each segment based on uncertainty estimates from previously determined segments. This stepwise progression inherently prevents averaging across variable segment positions, as later segments depend on the cumulative trajectory of earlier decisions. The algorithmic constraint ensures temporal consistency in segmentation rather than post hoc aggregation.
> > >
> > > **Q5:**
> > >
> > > A5: We would like to point out that we follow the recommended settings from prior work to set the hyperparameters of baselines [1, 2, 4]. The values of baselines are different for Llama and Mistral. The grid search is mainly conducted for our proposed CARDS, which is a standard practice in previous papers [1, 2, 3].
> > >
> > > [1] ARGS: Alignment as Reward-Guided Search. ICLR 2024.
> > >
> > > [2] TreeBoN: Enhancing Inference-Time Alignment with Speculative Tree-Search and Best-of-N Sampling. 2024.
> > >
> > > [3] Reward-Augmented Decoding: Efficient Controlled Text Generation With a Unidirectional Reward Model. EMNLP 2023.
> > >
> > > [4] RAIN: Your Language Models Can Align Themselves without Finetuning. ICLR 2024.
> > >
> > > **Q8:**
> > >
> > > A8: We kindly request the reviewer’s advice on how to compute the standard deviation without incurring additional costs. To our understanding, computing standard deviation requires repeating experiments multiple times, which is often infeasible in LLM experiments.
> > >
> > > Regarding the win/tie/lose results between CARDS and PPO: PPO is an RL-based method that is significantly more resource-intensive compared to test-time methods. The fact that CARDS achieves a competitive win-tie record against PPO already demonstrates its strong alignment quality.
> > >
> > > **Q10:**
> > >
> > > A10: We have tried to modify these unrelated hyperparameters, but found that further accelerating inference is not possible without compromising alignment quality. Inference speed is primarily determined by the segment length, which represents a fundamental distinction between uncertainty-based segmentation and sentence segmentation.

---

> ### Author Response · Authors · 2025-06-08
>
> Dear Reviewer,
>
> Thank you for your valuable feedback and engagement throughout the review process. We want to remind you that the deadline for the author-reviewer discussion period is approaching.
>
> We have not yet received a response to our posted replies, which we believe address the concerns you raised regarding our paper. We kindly encourage you to review our responses and, if you find them satisfactory, to reconsider your rating.
>
> Authors

---

### Comment · Area_Chair_3Q1m · 2025-06-05
**Reviewers, please engage**

Requesting the reviewers to read all reviews and responses and consider the arguments presented so that we can move closer to consensus on this paper.

Thank you! Area Chair

---

### Decision · Program_Chairs · 2025-07-08

**Decision:**

Accept

**Comment:**

The paper proposes CARDS, a decoding-time alignment algorithm for large language models that generates text in small segments using rejection sampling. It segments the output dynamically based on uncertainty, stopping sampling before high-entropy predictions, and evaluates each segment with a reward model to decide acceptance. Accepted segments are appended to the context for the next round of sampling. This method aims to improve efficiency by reducing redundant calls to both the language model and the reward model, compared to existing approaches. The authors demonstrate that segment-level rewards correlate well with full-sequence rewards and show empirical improvements on several standard alignment benchmarks.

Pros:
- Proposes a simple and effective segment-level rejection sampling method
- Uses uncertainty-based segmentation aligned with LM internal dynamics
- Demonstrates improved efficiency (around 70% decoding speed-up)
- Shows strong empirical performance outperforming multiple baselines
- Some reviewers say that the work provides comprehensive experiments and ablation studies

Cons:
- Insufficient experimental detail and unclear hyperparameter settings
- Some confusion or inconsistency in evaluation metrics and reported results

The reviews surfaced enough points of confusion that the authors are strongly encouraged to revise the paper for clarity.  Since much information is included in appendices, it would be helpful to include forward references to these throughout the paper so that readers with deeper questions can easily find the details.